# Implicit Kernel Meta-Learning Using Kernel Integral Forms

John Isak Texas Falk[1,2]          Carlo Ciliberto[1]          Massimiliano Pontil[1,2]

[1]Dept. of Computer Science, University College London, U.K.
[2]CSML, Italian Institute of Technology, Genoa, Italy

## Abstract

Meta-learning algorithms have made significant progress in the context of meta-learning for image classification but less attention has been given to the regression setting. In this paper we propose to learn the probability distribution representing a random feature kernel that we wish to use within kernel ridge regression (KRR). We introduce two instances of this meta-learning framework, learning a neural network pushforward for a translation-invariant kernel and an affine pushforward for a neural network random feature kernel, both mapping from a Gaussian latent distribution. We learn the parameters of the pushforward by minimizing a meta-loss associated to the KRR objective. Since the resulting kernel does not admit an analytical form, we adopt a random feature sampling approach to approximate it. We call the resulting method Implicit Kernel Meta-Learning (IKML). We derive a meta-learning bound for IKML, which shows the role played by the number of tasks $T$, the task sample size $n$, and the number of random features $M$. In particular the bound implies that $M$ can be the chosen independently of $T$ and only mildly dependent on $n$. We introduce one synthetic and two real-world meta-learning regression benchmark datasets. Experiments on these datasets show that IKML performs best or close to best when compared against competitive meta-learning methods.

## 1 INTRODUCTION

A common scenario in the real world is learning from similar tasks in order to transfer this knowledge to new tasks. In machine learning this setting is called meta-learning or learning-to-learn [Baxter, 2000, Thrun and Pratt, 1998],

where we assume that a set of tasks are sampled from a meta-distribution on supervised learning problems. The goal is to design *meta-algorithms* which from a set of tasks output a learning algorithm. This algorithm should perform well on average with respect to the tasks sampled from the meta-distribution, analogous to having low risk in supervised learning.

Meta-learning operates on top of an inner algorithm, tuning it to perform better on new tasks. The meta-algorithm acts at an outer level, relying on the inner algorithm to compute a meta-loss and corresponding meta-gradient, based on which a meta-parameter associated to the inner algorithm is updated [see e.g. Franceschi et al., 2018]. For example, in regression settings, a common choice of inner algorithm is ridge regression and the meta-parameter is a representation or embedding shared across the tasks that we wish to meta-learn [Bertinetto et al., 2018].

There has been considerable interest in meta-learning for few-shot image classification [Finn et al., 2017, Koch et al., 2015, Li et al., 2017, Ren et al., 2018, Rusu et al., 2018, Snell et al., 2017, Vinyals et al., 2016] but less attention has been given to designing meta-learning algorithms for regression. Most few-shot regression benchmarks fall under that of interpolating sinusoidals or a variety thereof [Finn et al., 2017, 2018, Oreshkin et al., 2018] which lacks many aspects of real-world regression problems such as being multivariate and noisy. This highlights the importance of more realistic meta-learning regression datasets and how to design meta-learning algorithms in this setting. In this paper we aim to close this gap.

Meta-learning algorithms employ a variety of different kinds of base algorithms, ranging from metric based to optimization based, and to black-box ones. A common theme is to learn a shared representation which lead to faster adaptation of a base learning algorithm to new tasks. Often the representation is modeled by a neural network. Indeed, recently [Raghu et al., 2019, Tian et al., 2020] observed that the representation is the most important part of meta-learning

*Accepted for the 38th Conference on Uncertainty in Artificial Intelligence* (UAI 2022).

algorithms.

In this paper we extend this thinking further, in that we implicitly learn the representation via a kernel function from a large class of kernels defined by a random feature form. This kernel is in turn implicitly parametrized by a neural network pushforward which is learned by a meta-algorithm. When using the random feature family of translation-invariant kernels this has two main advantages: since kernel algorithms can be expressed in terms of inner products of features which are simple to compute we don't have to work with this high-dimensional feature space directly. We show that modeling the kernel directly leads to improved performance in the meta-learning regression case. A second advantage is that translation invariant kernels might be used as "plug-in" representations. We also experiment with using a neural network random feature representation, effectively combining ensembling with with random features.

**Contributions** The principal contribution of this paper is a method for meta-learning regression together with a bound on the excess risk which highlights how problem-specific quantities impact the number of random features needed to generalize. In particular, our method can be used to learn within a family of translation invariant kernels that is well-suited when using kernel ridge regression as the class of base learning algorithms. According to Bochner's Theorem [see e.g. Rahimi et al., 2007], these kernels are parameterized by a distribution in the frequency space. In line with [Li et al., 2019], we parametrize this distribution as a neural network pushforward. The weights of the network are learned from a sequence of datasets within a meta-learning setting. Although we focus on distributions in the context of Bochner's theorem, our framework extends directly to radial kernels using Schoenberg's theorem [Schoenberg, 1938]. Additionally we experiment with using a neural network random feature kernel, an extension of R2D2 [Bertinetto et al., 2018], and show competitive performance.

Finally, we introduce three novel meta-learning regression benchmark datasets, one synthetic and two real-world and show that our algorithm ranks at the top or close to competing meta-learning regression algorithms. We believe these results, including the theoretical guarantees together with the flexibility and ease of our method, make it a competitive candidate to be used as a plug-in meta-learning algorithm in general contexts.

**Related Work** Learning-to-learn or meta-learning can be traced back to at least [Schmidhuber, 1987] with one seminal work being [Baxter, 2000]. Well-developed theory exists in the batch case [Maurer, 2005, Maurer et al., 2016] and lately similar results have been developed in the online setting [Balcan et al., 2019, Denevi et al., 2019].

Recent advances in the image few-shot classification setting [Fei-Fei et al., 2006, Lake et al., 2011] starting with the work of [Finn et al., 2017, Snell et al., 2017, Vinyals

et al., 2016] has lead to renewed interest in meta-learning, notably from the deep learning community by formulating it as an optimization problem [Ravi and Larochelle, 2017]. While classification has received a lot of interest, regression has been given less attention. Some examples are given by [Patacchiola et al., 2020, Titsias et al., 2020, Tossou et al., 2019] who apply gaussian processes [Williams and Rasmussen, 2006] together with deep kernel learning [Wilson et al., 2016] to regression. From the ridge regression point of view; Kong et al. [2020] investigate theoretically the meta-mixed linear regression setting while Nguyen et al. [2021] applied kernel ridge regression (KRR) to meta-learn dataset compression.

Our work can be traced directly to ideas from [Li et al., 2019, Sinha and Duchi, 2016, Zhen et al., 2020] to leverage the characterization provided by Bochner's theorem for kernel learning [Cristianini et al., 2006, Ong et al., 2005]. In [Sinha and Duchi, 2016] they fine-tune a convex combination of sampled kernels in a supervised learning setting using kernel target alignment [Cristianini et al., 2006]. We also mention the work [Zhen et al., 2020] which apply variational inference to optimize a latent variable model for few-shot learning, and [Li et al., 2019] where they learn an implicit kernel using a pushforward in the case that the learning objective is linear in the kernel evaluations.

**Organization** Sec. 2 introduces the meta-learning setting. We describe our proposed method in Sec. 3, analyze it in Sec. 4 and benchmark it in Sec. 5. We discuss our findings in Sec. 6.

# 2 META-LEARNING PROBLEM

In this section we introduce the main elements of the meta-learning setting and introduce the notion of stochastic meta-learning algorithm. To this end, we first recall the standard notion of supervised learning problem.

**Supervised Learning** Given an input $\mathcal{X}$ and output $\mathcal{Y}$ set, a supervised learning problem is characterized by a data generating distribution $\mu \in \mathcal{P}(\mathcal{X} \times \mathcal{Y})$ on the joint space $\mathcal{X} \times \mathcal{Y}$ and a loss function $\ell : \mathcal{Y} \times \mathcal{Y} \to \mathbb{R}$ measuring prediction errors. The goal of a supervised learning problem is to find a map $f : \mathcal{X} \to \mathcal{Y}$ minimizing the *risk*

$$\mathcal{R}_\mu(f) = \mathbb{E}_{(x,y) \sim \mu}\, \ell(f(x), y). \tag{1}$$

In practice, the data generating distribution is unknown and only a finite number $n_{\mathrm{tr}}$ of examples $D^{\mathrm{tr}} = (x_i, y_i)_{i=1}^{n_{\mathrm{tr}}}$ independently sampled from $\mu$ are available (denoted here by $D^{\mathrm{tr}} \sim \mu^{n_{\mathrm{tr}}}$). A learning algorithm is a function mapping datasets into candidate solutions to (1).

**Learning Algorithm** Let $\mathcal{Y}^{\mathcal{X}} = \{f : \mathcal{X} \to \mathcal{Y}\}$ be the space of all functions from $\mathcal{X}$ to $\mathcal{Y}$ and $\mathcal{D}$ the space of all datasets of any size on $\mathcal{X} \times \mathcal{Y}$. Then, a learning algorithm (referred to as *inner algorithm* in meta-learning) is

a function $A : \mathcal{D} \to \mathcal{Y}^{\mathcal{X}}$, mapping datasets $D \in \mathcal{D}$ to functions $f : \mathcal{X} \to \mathcal{Y}$. Typically, learning algorithms are parametrized as $A(\cdot) = A(\theta, \cdot)$, by a set of so-called hyper-parameters (here referred to as *meta-parameters*) $\theta \in \Theta$, that allow to adapt the algorithm to the specific problem. Typical examples of hyperparameters include the regularizer in Tikhonov regularization or the number of iterations of an early-stopping procedure. Ideally, we aim to find the best meta-parameter for a given task, namely the $\theta$ minimizing the expected risk $\mathcal{R}(A(\theta, D))$. We do not have access to $\mu$ but we can sample a *validation* set $D^{\mathrm{val}} \sim \mu^{n_{\mathrm{val}}}$ and consider the empirical risk

$$\hat{\mathcal{R}}(f, D^{\mathrm{val}}) = \frac{1}{n_{\mathrm{val}}} \sum_{(x,y) \in D^{\mathrm{val}}} \ell(f(x), y), \qquad (2)$$

as a suitable proxy. Since $D^{\mathrm{val}}$ and $D^{\mathrm{tr}}$ are sampled independently, $\hat{\mathcal{R}}(A(\theta, D^{\mathrm{tr}}), D^{\mathrm{val}})$ is an unbiased estimator of $\mathcal{R}_\mu(A(\theta, D^{\mathrm{tr}}))$. Given a train and validation set $D = (D^{\mathrm{tr}}, D^{\mathrm{val}})$, the process of minimizing the meta-loss

$$L(\theta, D) = \hat{\mathcal{R}}(A(\theta, D^{\mathrm{tr}}), D^{\mathrm{val}}), \qquad (3)$$

with respect to $\theta$ is known as *cross-validation*.

**Meta-Learning** The meta-learning paradigm lifts the notion of cross-validation to the level of multiple tasks: assuming that we have access to many supervised learning problems (or tasks) sharing some form of similarity, meta-learning aims to find a single set of meta-parameters $\theta$ that works well across all tasks. More formally, we assume that the tasks are sampled from a *meta-distribution* $\rho$. From each $\mu \in \mathcal{P}(\mathcal{X} \times \mathcal{Y})$ sampled from $\rho$, we then sample a pair of datasets $D = (D^{\mathrm{tr}}, D^{\mathrm{val}}) \sim \mu^n$ with $n = n_{\mathrm{tr}} + n_{\mathrm{val}}$ (even though in the following we assume $n_{\mathrm{tr}}$ and $n_{\mathrm{val}}$ to be fixed for simplicity, our discussion can be extended to more general settings). Then, meta-learning is formulated as the problem of finding the meta-parameters $\theta \in \Theta$ minimizing the *transfer risk* [Denevi et al., 2018]

$$\mathcal{E}(\theta) = \mathbb{E}_{\mu \sim \rho} \mathbb{E}_{D \sim \mu^n} L(\theta, D). \qquad (4)$$

If $L(\cdot, D)$ is (sub)differentiable, we can adopt standard stochastic first order method (e.g. SGD or Adam [Kingma and Ba, 2015]) to approximate the optimal meta-parameters. This consists in iteratively sampling a task $\mu_t \sim \rho$ and a train-val split $D_t \sim \mu_t^n$ at each time step $t = 1, \ldots, T$. Then, update the meta-parameters, e.g. via the SGD rule $\theta_{t+1} = \theta_t - \eta \nabla_\theta L(\theta, D_t)$. We refer to Alg. 1 for a concrete example in the setting discussed in this work.

**Meta Representation Learning** In practice, the above approach might pose computational challenges since, by the chain rule, differentiating $L$ requires computing $\nabla_\theta A(\theta, D)$. Depending on the inner algorithm $A$, its gradient with respect to the meta-parameters $\theta$ might be hard to compute or not even exist. In the literature, a wide range of meta-learning strategies have been proposed, considering different choices of inner algorithm $A$ and meta-parameters $\theta$. For example, [Bertinetto et al., 2018] considered the case that $A$ performs ridge regression (see Sec. 3.1) and $\theta$ parameterizes the weights of a feature map $\phi_\theta : \mathcal{X} \to \mathbb{R}^d$ (e.g. a neural network). Leveraging the closed-form solution of the ridge regression estimator, this allows us to efficiently compute the gradient $\nabla_\theta A(\theta, D)$. In settings where $A$ is minimizing the empirical risk but with a loss function that does not admit a closed form, we can adopt a bi-level optimization perspective [Franceschi et al., 2018]. This amounts to interpret $A$ as returning the $T$-th iteration of an iterative optimization algorithm. This allows to access $\nabla_\theta A(\theta, D)$ by recursively differentiating along the iterates. This approach is related to the well-known MAML algorithm [Finn et al., 2017], which proposed to perform fine-tuning of a shared starting network $f_\theta : \mathcal{X} \to \mathcal{Y}$ with weights $\theta$, that is adapted by $A(\theta, D) = f_{\theta'}$ to each new task by performing a step of gradient descent $\theta' = \theta - \eta \nabla_\theta \hat{\mathcal{R}}(\phi_\theta, D)$, to fit the training data. In the following we introduce the family of inner algorithms (and corresponding parameters) proposed in this work to tackle the meta-learning problem.

## 3 IMPLICIT KERNEL META-LEARNING

We now introduce the propose meta-learning strategy. While most previous work focused on learning a shared data representation or feature map [Bertinetto et al., 2018, Finn et al., 2017, Franceschi et al., 2018] across tasks, here we propose the dual approach of learning a shared kernel function.

### 3.1 REPRODUCING KERNELS AND FEATURE MAPS

Reproducing kernels are a well-established tool in machine learning, at the root of most non-parametric algorithms [Schölkopf and Smola, 2002]. They consist of positive definite functions $K : \mathcal{X} \times \mathcal{X} \to \mathbb{R}$ that may be interpreted as a similarity between data points. A fundamental result dating back to Moore and Aronszajn [see e.g. Aronszajn, 1950, Cucker and Smale, 2002, Schölkopf and Smola, 2002, and references therein] establishes that a kernel is into one-to-one correspondence with a (possibly infinite dimensional) Hilbert space $\mathcal{H}_K$ of real-valued functions on $\mathcal{X}$, such that for every $x \in \mathcal{X}$ and $f \in \mathcal{H}_K$, the function $K(x, \cdot) \in \mathcal{H}_K$ and $\langle f, K(x, \cdot) \rangle_K = f(x)$, where $\langle \cdot, \cdot \rangle_K$ denotes the inner product in $\mathcal{H}_K$. A kernel is in duality with the notion of feature map: given a mapping $\phi : \mathcal{X} \to \mathbb{H}$ into a Hilbert space $\mathbb{H}$ with inner product $\langle \cdot, \cdot \rangle$ such that $K_\phi(x, x') \equiv \langle \phi(x), \phi(x') \rangle$ is a reproducing kernel. The converse is also true, namely for any kernel $K$ there exists a Hilbert space $\mathbb{H}$ and feature map $\phi_K : \mathcal{X} \to \mathbb{H}$ such that $K(x, x') = \langle \phi_K(x), \phi_K(x') \rangle$ [Aronszajn, 1950]; when $\mathcal{X}$ is compact we can choose $\mathbb{H} = \ell_2$, the space of square summable sequences. A key practical advantage of

kernels is that they allow to learn functions parametrized as $f(x) = \langle f, \phi_K(x)\rangle$ even when $\mathcal{H}_K$ is infinite dimensional (namely the feature vector $\phi_K(x)$ has infinitely many entries). As a concrete example we recall the case of kernel ridge regression [see e.g Caponnetto and De Vito, 2007, Steinwart and Christmann, 2008] which we will use also as plug-in inner algorithm for the proposed meta-learning approach in this work.

**Kernel Ridge Regression** Kernel ridge regression (KRR) performs Tikhonov regularization using the least-square loss function over the space of hypotheses associated to a reproducing kernel [see e.g. Schölkopf and Smola, 2002]. More precisely, assume $\mathcal{Y} \subset \mathbb{R}$. Given a dataset $D^{\mathrm{tr}} = (x_i, y_i)$, and a kernel function $K : \mathcal{X} \times \mathcal{X} \to \mathbb{R}$, KRR is the algorithm

$$A_{\mathrm{KRR}}(K, D^{\mathrm{tr}}) = \underset{f \in \mathcal{H}_K}{\operatorname{argmin}} \; \hat{\mathcal{R}}(f, D^{\mathrm{tr}}) + \lambda\|f\|_K^2, \quad (5)$$

with $\lambda > 0$ a regularization parameter. Thanks to the reproducing property of the kernel, (5) can be solved in closed form. We have that for any $x \in \mathcal{X}$, that

$$A_{\mathrm{KRR}}(K, D^{\mathrm{tr}})(x) = \sum_{i=1}^{n} \alpha_i K(x_i, x) \qquad (6)$$

with $\alpha = (G + \lambda n I)^{-1} y$, where $G = (K(x_i, x_j))_{i,j=1}^{n_{\mathrm{tr}}}$ is the $n_{\mathrm{tr}} \times n_{\mathrm{tr}}$ kernel (Gram) matrix, $I$ the $d \times d$ identity matrix, and with some abuse of notation we let $y = (y_1, \ldots, y_{n_{\mathrm{tr}}})^\top \in \mathbb{R}^{n_{\mathrm{tr}}}$ be the vector of output examples. Notice that we highlighted the dependency of KRR with respect to the kernel $K$. This suggests that in meta-learning settings one might be interested in learning the kernel as a meta-parameter.

## 3.2 LEARNING TRANSLATION INVARIANT KERNELS

The definition of positive definite function underlying the notion of reproducing kernel is very general. Therefore, to formulate the problem of meta-learning a kernel, we need first to identify a suitable family. In [Rudi and Rosasco, 2017] they introduce a "recipe" for random feature kernels defined by a random feature map $\varphi : \mathcal{X} \times \Omega \to \mathbb{R}^o$ and a distribution $\tau$ so that any kernel in this family has the form

$$K(x, x') = \int_\Omega \varphi(x, \omega)^\top \varphi(x, \omega) \mathrm{d}\tau(\omega). \qquad (7)$$

Given the focus of this work towards regression settings, we first consider the class of *translation invariant* kernels, which are particularly suited to deal with such settings and are *interpretable* ( see e.g. Fig. 2 in the appendix). Let $\mathcal{X} = \mathbb{R}^d$. A kernel $K$ is called *translation invariant* if $K(x, x') = g(x - x')$ for some function $g : \mathbb{R}^d \to \mathbb{R}$; a well-known example is the Gaussian $K(x, x') = e^{-\|x-x'\|^2/\sigma^2}$

with $\sigma > 0$. A famous theorem by Bochner [see e.g. Rahimi et al., 2007, Rudin, 1962, Sriperumbudur and Szabo, 2015], adapted here to real-valued kernels, establishes that any properly re-scaled continuous bounded translation invariant function $K : \mathbb{R}^d \times \mathbb{R}^d \to \mathbb{R}$ is a kernel if and only if there exists a probability measure $\tau \in \mathcal{P}(\mathbb{R}^d)$ such that

$$K(x, x') = K_\tau(x, x') \equiv \int \cos(\langle \omega, x - x' \rangle) \mathrm{d}\tau(\omega), \quad (8)$$

which can be written in the form of (7) by expanding the cosine using the trigonometric identity $\cos(x - y) = \cos(x)\cos(y) + \sin(x)\sin(y)$. We call any kernel that can be written in the form of (7) a *Bochner kernel*. Eq. (8) implies that we can represent the class of translation invariant kernels as $\mathcal{T} = \{K_\tau \mid \tau \in \mathcal{P}(\mathbb{R}^d)\}$. Thus we can translate the problem of learning a kernel to that of learning a probability distribution. This perspective is in line with the implicit kernel learning approach devised in [Li et al., 2019] for generative modeling and single task settings. The second type of kernel is inspired by the success of using neural network to extract features and is given by letting $\varphi(x, \omega)$ be a neural network with $\omega$ the weights and $\tau$ a distribution over $\omega$.

**Pushforward Models** To learn the underlying distribution $\tau$ we consider a parametrization in terms of a pushforward model. More formally, let $\mathcal{N}$ be the unit Gaussian distribution over a latent space $\mathcal{Z}$ and let $\psi_\theta : \mathcal{Z} \to \mathbb{R}^d$ be a vector-valued function parameterized by a vector $\theta \in \Theta$ (e.g. a neural network with weights $\theta$). We denote by $\tau_\theta = \psi_\theta \# \mathcal{N}$ the probability distribution such that, the process of sampling $\omega \sim \tau_\theta$ is equivalent to first sampling $z \sim \mathcal{N}$ and then taking $\psi_\theta(z) = \omega$.[1] This is the strategy adopted to model the generator distribution in generative adversarial networks (GAN) settings and in the implicit kernel learning approach of [Li et al., 2019]. Several alternatives for the latent distribution $\mathcal{N}$ are possible (e.g. uniform). Under the notation above, we adopt as inner algorithm,

$$A(\theta, D) = A_{\mathrm{KRR}}(K_{\tau_\theta}, D), \qquad (9)$$

namely KRR trained with a translational invariant kernel $K_{\tau_\theta}$ meta-parametrized by the pushforward map $\tau_\theta$. Below we give an example where a parametrization of $\tau_\theta$ yields an analytic form for the corresponding $K_{\tau_\theta}$; see the appendix for a derivation.

**Example 1** (Affine Pushforward Maps). *Let $\theta = (Q, b)$ with $Q \in \mathbb{R}^{d \times d}$ and $b \in \mathbb{R}^d$ and consider the affine pushforward map $\psi_{(Q,b)}(s) = Qs + b$. In these settings, the kernel $K_{\tau_{(Q,b)}}$ can be expressed analytically as*

$$K_{\tau_{(Q,b)}}(x, x') = \cos(\langle b, x - x' \rangle) e^{-\|Q^\top(x-x')\|^2/2}. \quad (10)$$

The example above identifies a relevant family of kernels that are particularly amenable for meta-learning. Thanks

---

[1] Formally, for any $B \subseteq \mathbb{R}^d$, $\tau_\theta(B) = \mathcal{N}(\{z \mid \psi_\theta(z) \in B\})$.

to the analytic form of affine pushforward kernels, we can easily compute meta-gradients and thus directly minimize the transfer risk $\mathcal{E}(\theta)$. On the other hand, if we consider more expressive maps $\psi_\theta$, we will hardly be able to obtain $K_{\tau_\theta}$ in analytic form. Still, this may be well worth the effort: while for large training sets the difference between (10) and a more sophisticate kernel may be less severe since any universal kernel is optimal [Caponnetto and De Vito, 2007], in the few-shot learning setting (where we have small training sets) the inductive bias plays an important role and being able to modify the kernel in a flexible way is key.

### 3.3 STOCHASTIC META-LEARNING

The discussion above highlighted that except for a few special cases (see e.g. Example 1), given a distribution $\tau_\theta$ it is not possible to compute the kernel $K_{\tau_\theta}$ (and its gradient with respect to $\theta$) analytically. In principle, this might prevent us from applying meta-learning algorithms of the form in (9). To circumvent this issue, we consider a strategy based on random features [Rahimi et al., 2007, Rudi and Rosasco, 2017]. Rather than evaluating $K_{\tau_\theta}$, we sample a set $S = (s_j)_{j=1}^M$ from $\mathcal{N}$ and then approximate the ideal Bochner kernel by the *random features kernel*

$$K_{\hat\tau_{\theta S}}(x, x') = \frac{1}{M} \sum_{j=1}^M \cos(\langle \psi_\theta(s_j), x - x' \rangle), \quad (11)$$

where $\hat\tau_{\theta S} = \frac{1}{M} \sum_{j=1}^M \delta_{\psi_\theta(s_j)}$ is an empirical distribution associated to $\tau_\theta$ and $\delta_\omega$ denotes a Dirac's delta centered in $\omega \in \mathbb{R}^d$ which we call *frequency*. Thanks to the characterization of $K_{\tau_\theta}$ as an expectation in (8), we have that

$$K_{\tau_\theta}(x, x') = \mathbb{E}_{S \sim \mathcal{N}^M} K_{\hat\tau_{\theta S}}(x, x'), \quad (12)$$

namely $K_{\hat\tau_{\theta S}}$ is an unbiased estimator of $K_{\tau_\theta}$. It is possible to prove also non-asymptotic results bounding the distance beetween the two kernels in sup norm [Rahimi et al., 2007].

**Stochastic Meta-Learning** We now introduce a stochastic variant to the meta-learning approach from Sec. 2, by defining the meta-loss associated to a set of random features

$$L(\theta, S, D) = \hat{\mathcal{R}}(A_{\mathrm{KRR}}(K_{\tau_\theta S}, D^{\mathrm{tr}}), D^{\mathrm{val}}), \quad (13)$$

and the corresponding transfer risk

$$\mathcal{E}_M(\theta) = \mathbb{E}_{\mu \sim \rho} \mathbb{E}_{D \sim \mu^n} \mathbb{E}_{S \sim \mathcal{N}^M} L(\theta, S, D), \quad (14)$$

which we will also denote $\mathcal{E}(\theta, S)$ when wanting to highlight the dependence on $S$ explicitly.

In this work we propose to address the stochastic meta-learning problem

$$\min_{\theta \in \Theta} \mathcal{E}_M(\theta). \quad (15)$$

Alg. 1 provides the pseudocode for a (meta) stochastic gradient descent algorithm applied to this problem. At each iteration $t = 1, \ldots, T$, we sample a new task $\mu_t$ and datasets

---

**Algorithm 1** Implicit Kernel Meta-Learning

**Input:** meta-distribution $\rho$, step-sizes $(\gamma_t)_{t=1}^\infty$, number of random features $M$, initial meta-parameters $\theta_0$, total number of iterations $T$.
**For** $t = 1, \ldots, T$
    Sample a task/dataset $D = (D^{\mathrm{tr}}, D^{\mathrm{val}})$ from $\rho$
    Sample $M$ random features $S$ from $\mathcal{N}$
    Form $K_{\tau_{\theta_t S}}$ and compute $L(S, \theta_t, D)$ as in (13)
    Get $\nabla_\theta L(\theta_t, S, D) = \mathrm{AUTOGRAD}(L(\cdot, S, D), \theta_t)$
    Update $\theta_{t+1} \leftarrow \theta_t - \gamma_t \nabla L(\theta_t, S, D)$
**Return** $\theta_T$

---

$D_t = (D_t^{\mathrm{tr}}, D_t^{\mathrm{val}}) \sim \mu_t^n$ and a set of random features $S_t \sim \mathcal{N}^M$ and then perform a gradient descent step in the direction of $\nabla_\theta L(\theta_t, S_t, D_t)$. Note that the gradient can be computed by means of automatic differentiation (AUTOGRAD) [see e.g. Baydin et al., 2018]. Many other strategies to perform this optimization step are available, such as Adam [Kingma and Ba, 2015]. While using a large number of random feature may seem expensive, both training and prediction time is linear in $M$, see the section on computational complexity in the appendix. We refer to this method as *Implicit Kernel Meta-Learning (IKML)*.

## 4 GENERALIZATION BOUND

We now study the generalization ability of the proposed meta-learning method. In particular, our goal is to study the effect of the number of random features on the performance of the meta algorithm. To present our observations we focus for simplicity on the case that the meta loss uses the task dataset for both training and validation, that is we use the empirical risk

$$\tilde{L}(\theta, S, D) = \hat{\mathcal{R}}(A_{\mathrm{KRR}}(K_{\hat\tau_{\theta S}}, D), D) \quad (16)$$

which is the empirical error of KRR with kernel (11) on the dataset $D$ instead of (13). For a collection of datasets $(D_t)_{t=1}^T$ and a sample $S = (s_j)_{j=1}^M$ from $\mathcal{N}$, define the multitask empirical risk

$$\hat{\mathcal{E}}_T(\theta, S) = \frac{1}{T} \sum_{t=1}^T \tilde{L}(\theta, S, D_t). \quad (17)$$

We aim to bound the excess transfer risk

$$\mathcal{E}_M(\hat\theta) - \mathcal{E}(\theta^*) \quad (18)$$

where $\theta^* \in \Theta$ is such that $\mathcal{E}(\theta^*) = \min_\theta \mathcal{E}(\theta)$ and $\hat\theta$ is the minimizer of the multitask empirical risk, which we call the multitask empirical risk minimizer (MERM) which in practice we approximate by the solution returned by Alg. 1.

**Theorem 1.** *Assume that $\mathcal{Z} = \mathcal{X} \times \mathcal{Y} \subseteq \mathbb{R}^d \times [0, 1]$, $\rho$ is a meta-distribution on $\mathcal{Z}$, the loss $\ell(y, \hat{y}) = (y - \hat{y})^2$ and*

*kernel family $\mathcal{K} = \{K_{\tau_\theta} \mid \theta \in \Theta\}$ is a family of Bochner kernels parameterized by some latent distribution $\mathcal{N}$ with support on $\mathbb{R}^l$ and a family of measurable functions $\{\psi_\theta : \mathbb{R}^l \to \mathbb{R}^d \mid \theta \in \Theta\}$. For any $n, M, T \in \mathbb{N}$ let the training task datasets $D_1, \ldots, D_T$ be given by iteratively sampling a task $\mu_t \sim \rho$ and $D_t \sim \mu_t^n$ and $S \sim \mathcal{N}^M$, the family of inner algorithms being KRR with kernels $K_{\tau_\theta} \in \mathcal{K}$ and fixed regularization parameter $\lambda > 0$ and $\hat{\theta}$ being the MERM over the task datasets and random features. Then, for $\delta \in (0, 1)$, with probability at least $1 - \delta$ over the datasets and random features*

$$\mathcal{E}_M(\hat{\theta}) - \mathcal{E}(\theta^*) \le O\left(\frac{\sqrt{M}R_{n,M,T}}{T\lambda\sqrt{n}} + \sqrt{\frac{\log\frac{1}{\delta}}{T}}\right) + \quad (19)$$

$$O\left(\frac{1}{\lambda\sqrt{n}}\right) + \quad (20)$$

$$O\left(\frac{1}{\sqrt{M}\lambda^3}\left(1 + \sqrt{\frac{G_n^* \log n}{\lambda^2 n}}\right)\right) \quad (21)$$

*where*

$$R_{n,M,T} = \mathbb{E}_{(D_t)_{t=1}^T \sim \hat{\rho}^T}\mathbb{E}_{S,\epsilon} \sup_{\theta \in \Theta} \sum_{i,j,t}^{n,M,T} \epsilon_{i,j,t}\langle \psi_\theta(s_j), x_i^t\rangle,$$
(22)

*the random variables $\epsilon_{i,j,t}$ being i.i.d Rademacher and $D \sim \hat{\rho}$ means first sampling $\mu \sim \rho$ and then $D \sim \mu^n$, and $G_n^* = \mathbb{E}_{\mu\sim\rho}\mathbb{E}_{D\sim\mu^n}\|(K_{\theta^*}(x_i, x_j))_{i,j=1}^n\|_\infty$.*

*Proof Sketch.* We discuss the key elements of the proof and present the full details in the appendix. We write $\mathcal{E}_M(\hat{\theta}) - \mathcal{E}(\theta^*) = \mathbb{E}_S[\mathcal{E}_M(\hat{\theta}, S) - \mathcal{E}(\theta^*)]$ and decompose the term inside the expectation as

$$\underbrace{\mathcal{E}(\hat{\theta}, S) - \hat{\mathcal{E}}(\hat{\theta}, S)}_{(A)} + \underbrace{\hat{\mathcal{E}}(\hat{\theta}, S) - \hat{\mathcal{E}}_T(\hat{\theta}, S)}_{(B)} + \underbrace{\hat{\mathcal{E}}_T(\hat{\theta}, S) - \hat{\mathcal{E}}_T(\theta^*, S)}_{(C)}$$

$$+ \underbrace{\hat{\mathcal{E}}_T(\theta^*, S) - \hat{\mathcal{E}}(\theta^*, S)}_{(D)} + \underbrace{\hat{\mathcal{E}}(\theta^*, S) - \mathcal{E}(\theta^*, S)}_{(E)} + \underbrace{\mathcal{E}(\theta^*, S) - \mathcal{E}(\theta^*)}_{(F)}$$

where $\hat{\mathcal{E}}(\theta, S)$ and $\hat{\mathcal{E}}_T(\theta, S)$ are the average empirical error and the multitask empirical error, for the meta-parameter $\theta$ and random features $S$; – see the secion on the bound in the appendix. Bounding the terms (A) and (E) leads to (19) while bounding the terms (B) and (D) leads to (20). The term (C) is the optimization error and is negative if we can minimize the empirical risk objective. Finally the term (F) is bounded using [Tropp, 2019, Theorem 2.1] and auxiliary results presented in the appendix. □

We now comment on the implications of the above theorem. The first term in the r.h.s. of (19) contains the unnormalized Rademacher complexity $R_{n,M,T}$ of the set

$\{(\langle\psi_\theta(s_j), x_i^t\rangle)_{i,j,t=1}^{n,M,T} : \theta \in \Theta\} \subseteq \mathbb{R}^{n \times M \times T}$. This is a measure of the capacity of the RKHS's we consider as part of using the kernel family $\mathcal{K}$ and quantifies the kernel families ability to fit random noise. While this quantity requires a case by case analysis it is often of order $\sqrt{T}$. Since in meta-learning the number of tasks is very large this term is negligible in many practical scenarios. For example following the reasoning in [Oneto et al., 2020] we obtain that $R_{n,M,T} = O(\sqrt{nMT})$. The number of random features should then be chosen so that the quantity (21) is smaller than (20). $G_n^*$ represents the size of best RKHS needed to explain the data averaged over the possible datasets sampled from the environment. In some sense it represents the degrees of freedom of the best model $\theta^*$ given the meta-distribution. A direct computation gives the condition

$$M > O\left(\frac{n}{\lambda} + \frac{G_n^* \log n}{\lambda^3}\right).$$

Since $G_n^* \in [1, n]$, we conclude that the number of random features needed by the algorithm in order to be competitive with meta-learning without random feature approximation is *independent of the number of tasks* and only mildly dependent on $n$. For example, assuming $\lambda = 1/\sqrt{n}$ we obtain that $M = \Omega(n^{\frac{3}{2}} \log n)$ or $M = \Omega(n^{\frac{5}{2}} \log n)$ when $G_n^* = 1$ or $G_n^* = n$, respectively. The case that $G_n^* = O(n)$ requiring more random features corresponds to a low rank Gram matrix, meaning that the tasks are strongly related. This is however worth the effort since in this case the optimal risk $\mathcal{E}(\theta^*)$ we compare to will be very small, because the optimal low rank kernel makes learning very easy. Finally we note that $\lambda$ being in the denominator of all terms is an artifact due to comparing to the best KRR algorithm $\theta^*$ instead of the quantity $\mathcal{E}^* = \mathbb{E}_{\mu\sim\rho}\mathcal{R}_\mu(f_\mu)$ where $f_\mu = \mathbb{E}[y|\cdot]$ is the optimal predictor for the distribution $\mu$ [see Denevi et al., 2019, for a discussion].

## 5 EXPERIMENTAL RESULTS

We evaluate the performance of the proposed meta-learning strategy on both synthetic and real experiments against several baselines. We make all datasets and code available as a Github repository.[2]

### 5.1 SYNTHETIC MULTIVARIATE REGRESSION

For IKML to be effective in realistic meta-learning regression scenarios it is important that it can approximate nontrivial functions defined on $\mathbb{R}^d$ where $d \gg 1$. To investigate this we create a synthetic high-dimensional meta-learning regression setting where each task is sampled from an RKHS $\mathcal{H}$ with a "complicated" kernel $K^o$. In particular, we choose $K^o$ to be the kernel given by Bochner's theorem and a pushforward of a 3-layers Multi-Layer Perceptron (MLP) with

---

[2]https://github.com/IsakFalk/IKML

32 hidden units per layer, ReLU activation functions and a 16-dimensional latent Gaussian distribution. The network was initialized with weights given by the PyTorch [Paszke et al., 2019] default initialization scaled by 100. Since this kernel lacks an analytic form, we sample 10000 frequencies and use the random features kernel from (11) in its place. The tasks are generated from a distribution on $f \in \mathcal{H}$ and a marginal distribution on inputs fixed across all tasks. For each task we sample $n = n_{\text{tr}} + n_{\text{val}} = 50 + 50$ inputs $(x_i)_{i=1}^n$, a function $f$ and create the task $(x_i, f(x_i))_{i=1}^n$, for more details see Sec. 5 in the appendix.

We compare the following meta-learning algorithms:

*IKML.* Alg. 1 parameterizing the pushforward $\psi_\theta$ for the measure $\tau_\theta$ with a three-layer MLP with hidden dimension set to 32 and the dimension of the latent space $\mathcal{Z} = \mathbb{R}^{16}$. The number of random features is set to $M = 10^4$.

*Gaussian MKL meta-KRR (GMKL).* Multiple Kernel Learning (MKL) with KRR as inner algorithm. The meta-algorithm consists in learning the weights of a kernel $K = \sum_{j=1}^k \lambda_j K_j$ that is a convex combinations of Gaussian kernels $K_j(x, x') = \exp(-\frac{1}{2\sigma_j^2}\|x - x'\|^2)$ with lengthscale $\sigma_j$ taken from an log-equidistant grid from $10^{-3}$ to $10^3$. The meta-learning algorithms learns the weights $\lambda$ parameterized in terms of the vector $z \in \mathbb{R}^k$ as $\lambda_j = \frac{\exp(z_j)}{\sum_{i=1}^k \exp(z_i)}$.

*MAML [Finn et al., 2017].* Optimizing through inner gradient descent with MLP to learn a good initalization in the outer loop. We use a three-layer MLP with 32 hidden units and ReLU activation functions.

*R2D2 [Bertinetto et al., 2018].* Ridge regression as inner algorithm, learning a shared feature map in the outer loop. We use a three-layer MLP with 32 hidden units and ReLU activation functions.

*Oracle.* Running a separate instance of KRR on each task, with the same kernel $K^o$ used to generate the tasks, and finding $\lambda$ by cross validation on the test set.

## 5.2 REAL-WORLD DATA EXPERIMENTS

We evaluate the proposed approach on two new real world meta-learning regression datasets adapted to the meta-learning setting from the UCI repository [Dua and Graff, 2017]. Apart from IKML and Gaussian MKL meta-KRR, we used the following algorithms in our experiments: *LS Biased Regularization [Denevi et al., 2019] (LSBR).* Running linear ridge regression with biased regularization $\lambda\|f - \theta\|^2$ in the inner algorithm, learning the bias $\theta$ in the outer loop.

*ANP [Kim et al., 2019].* Learns to map datasets to stochastic processes over functions using neural networks to do meta-learning. Predictor is the conditional mean of the stochastic process.

*Gaussian Oracle KRR (GO).* Gaussian KRR addressing each task as a separate learning problem but cross-validating the kernel bandwidth and regularization parameters $\sigma^2$ and $\lambda$ on the average validation error directly on the meta-test set.

We chose the baselines from landmark papers in the few-shot learning (MAML, R2D2, ANP, LSBR) and multiple-kernel learning (GMKL, GO) literature applicable to regression. We think these are natural baselines to compare against.

For both meta-learning datasets, we run the algorithms above in an online fashion where we use a meta-batch of 4 tasks per iteration sampled from the meta-train set. For IKML we fix the number of random features to 20000 which is on the order of $\Omega(n^{5/2}\log(n))$ if we would have pooled the train and validation set of 25 datapoints to one train set of size 50. Note however that further experiments show that in practice we can get away with as little as 2500 random features while mainting performance. Every 250 steps we sample 1000 tasks from the meta-validation set and evaluated the average meta-loss for each algorithm and save the model parameters. After training we sample 3000 tasks from the meta-test set. For the meta-test evaluation, for all algorithms, we use the meta-parameters with the lowest meta-validation error and get the test performance for all algorithms. We measure performance in terms of the root mean square error (RMSE). This procedure was run 5 times over different random seeds in order to get learning curves and results on the meta-test set. Below we describe the datasets and comment on the empirical evidence.

**Air Quality** The Beijing Air Quality dataset [Zhang et al., 2017] is a time-series dataset measuring air-quality and meterological factors at 12 air-quality monitoring sites. The meterological data for each site is matched with the closest of available weather stations. The data was collected hourly and from the period March 1st, 2013 to February 28th, 2017. Further details in Sec. 4.1 in the appendix.

We generate a task of train and validation size $n_{\text{tr}}, n_{\text{val}}$ by randomly picking a station and picking a contiguous subsequence of size $n = n_{\text{tr}} + n_{\text{val}}$ at random from the split. We append the feature "$t$" which is the local order of data points and then randomly assign $n_{\text{tr}}$ of the $n$ points to the train set and the rest to the validation set. This can be seen as a reconstruction problem: given data from sensor of which some have failed, we want to infer the output given an input at some points in time. We choose to use $n_{\text{tr}} = n_{\text{val}} = 25$.

After experimenting we use the following configuration of the algorithms; For Gaussian MKL meta-KRR we use 20 Gaussian kernels with lengthscale sampled geometrically from 1 to $10^{12}$ and learn the coefficients and regularisation parameter using the same parameterization as in the synthetic experiment with Adam and a meta-learning rate of 0.001. For LS Biased Regularization we learn the bias and regularisation parameter using Adam with meta-learning

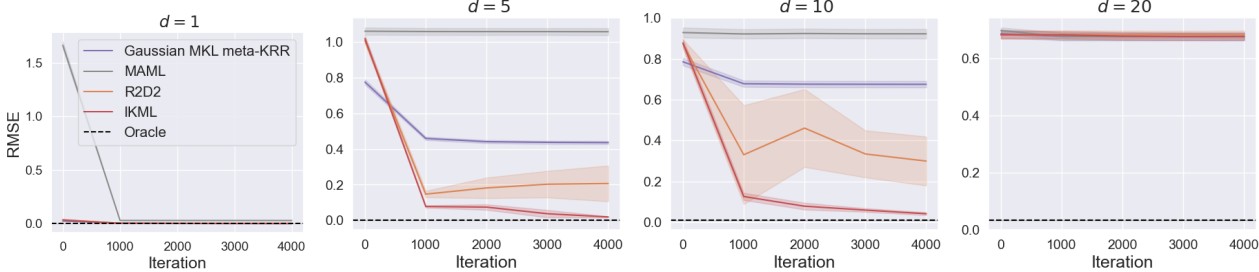

Figure 1: Learning curves of meta-test RMSE over three runs (mean $\pm$ 1 std) of Gaussian MKL meta-KRR , MAML, R2D2 and IKML together with the KRR Oracle on the synthetic meta-learning problem introduced in Sec. 5.1 for $d = 1, 5, 10, 20$. We generate $K^o$ once for each experiment and resample tasks for each run. Note that for low dimensions, MKL and R2D2 performs comparably to IKML. As the dimension increases, IKML outperforms all algorithms with performance on par with Oracle.

rate 0.01. We parameterized MAML with a 2-layer MLP with 64 hidden dimensions and with inner learning rate $10^{-7}$ and one adaptation step, learning the initialization using Adam with a meta-learning rate of 0.001. We found that using a very small inner learning rate and few steps was important to get MAML to converge. For R2D2, IKML and ANP we cross-validated to find the best set of hyparparameters, see Sec. 5 and Tab. 1 in the appendix for more information. For Gaussian meta-KRR we learn the lengthscale and regularisation parameter using Adam with a meta-learning rate of 0.001. We benchmark a neural network IKML, called IKML-MLP in where we use a 4-layer MLP with 64 hidden units, 8 output features and 500 random features trained using Adam with learning rate of $3 \cdot 10^{-4}$, see Sec. 5 in the appendix.

From Tab. 1 we can see that IKML performs best with R2D2 and IKML-MLP close seconds.

**Gas Sensor** The Gas Sensor Modulation dataset [Burgués et al., 2018] is a collection of multivariate timeseries collected in a controlled environment using MOX sensors for CO detection sampled at 3.5 Hz. Each task corresponds to a subsampled time-series from an experiment. As noted in [Burgués et al., 2018] the regression tasks are hard due to being heteroscedastic, non-normal and non-linear as a function of time but with tasks sharing a lot of structure, making it suitable as a meta-learning regression dataset. Further details in Sec. 4.2 in the appendix.

We benchmark the algorithms for $n_{\mathrm{tr}} = n_{\mathrm{val}} = 20$. After experimenting we use the following configuration of the algorithms; For Gaussian MKL meta-KRR we use 20 Gaussian kernels with lengthscale chosen geometrically from 1 to $10^8$ and learn the coefficients and regularisation parameter using the same parameterization as in the synthetic experiment with Adam and a meta-learning rate of 0.001. For LS Biased Regularization we learn the bias and regularisation parameter using Adam with meta-learning rate 0.01.

We parameterized MAML with a 4-layer MLP with 64

Table 1: Test RMSE on Beijing Air Quality and Gas Sensor. Best results in **bold**.

| Model | Air Quality RMSE | Gas Sensor RMSE |
|---|---|---|
| GMKL | $23.27 \pm 0.16$ | $9.61 \pm 0.07$ |
| LSBR | $21.68 \pm 0.29$ | $12.44 \pm 0.14$ |
| MAML | $34.96 \pm 3.58$ | $2.81 \pm 0.12$ |
| R2D2 | $20.23 \pm 0.55$ | $\mathbf{1.95 \pm 0.06}$ |
| Gaussian meta-KRR | $25.08 \pm 0.48$ | $9.80 \pm 0.09$ |
| GO | $25.94 \pm 0.91$ | $12.78 \pm 0.10$ |
| IKML | $\mathbf{19.14 \pm 0.93}$ | $2.80 \pm 0.10$ |
| IKML-MLP | $20.77 \pm 0.57$ | $2.06 \pm 0.09$ |
| ANP | $33.77 \pm 0.70$ | $2.12 \pm 0.09$ |

hidden units with inner learning rate $10^{-4}$ and one adaptation step, learning the initialization using Adam with meta-learning rate of $10^{-4}$. For R2D2, IKML and ANP we cross-validated to find the best set of hyparparameters, see Sec. 5 and Tab. 1 in the appendix for more information. For Gaussian meta-KRR we learn the lengthscale and regularization parameter using Adam with a meta-learning rate of 0.001. IKML-MLP is as for Gas Sensor, but with 2 layers and 100 random features.

As can be seen from the table, IKML and MAML gets a low meta-test error after R2D2 and IKML-MLP as can be seen in Tab. 1.

**Additional Metrics** For the algorithms R2D2, IKML and ANP, with the same setting and training strategy as outlined for Air Quality and Gas Sensor datasets, we evaluate them on two additional metrics: mean average error (MAE) and symmetric mean absolute scaled error (SMAPE)[3] [Chicco et al., 2021]. From Tab. 2 we see that IKML performs the best on all metrics on the Air Quality dataset with ANP performing poorly. For Gas Sensor R2D2 performs best

---

[3]Note that we present this as a ratio instead of as a percentage.

Table 2: Test RMSE / MAE / SMAPE on Beijing Air Quality and Gas Sensor datasets for R2D2, IKML and ANP.

| Model | Air Quality | | | Gas Sensor | | |
|---|---|---|---|---|---|---|
| | RMSE | MAE | SMAPE | RMSE | MAE | SMAPE |
| R2D2 | $20.23 \pm 0.55$ | $11.67 \pm 0.40$ | $0.24 \pm 0.01$ | $\mathbf{1.95 \pm 0.06}$ | $\mathbf{0.94 \pm 0.09}$ | $0.18 \pm 0.05$ |
| IKML | $\mathbf{19.14 \pm 0.93}$ | $\mathbf{10.62 \pm 0.19}$ | $\mathbf{0.22 \pm 0.00}$ | $2.80 \pm 0.10$ | $1.61 \pm 0.26$ | $0.24 \pm 0.03$ |
| ANP | $33.77 \pm 0.70$ | $21.08 \pm 0.40$ | $0.35 \pm 0.01$ | $2.12 \pm 0.09$ | $1.06 \pm 0.06$ | $\mathbf{0.09 \pm 0.01}$ |

except for SMAPE where ANP performs much better.

# 6 CONCLUSION AND FUTURE WORK

We introduced a framework for implicit kernel meta-learning (IKML) in context of translation-invariant and deep random kernel families. Our approach focuses on problems where data does not present a clear input structure (in contrast e.g. to image classification settings) and using a plug-in translation invariant kernel might be a safer strategy. Our approach leverages the characterization of random feature kernels, in particular the translation invariant kernels granted by Bochner's theorem and ideas from the random features literature to learn it in practice. We derive a novel bound on the excess transfer risk shedding light on how to choose the number of random features. To validate our method we introduced two real-world meta-learning regression datasets.

IKML achieve best or close-to-best performance on all of the datasets against state-of-the-art methods designed for few-shot image classification. We hypothesize that when the data does not have enough structure (e.g. in most regression settings), learning a deep representation – as done by state-of-the-art methods such as MAML or R2D2 – may be less effective. We leave further investigation to this question to future work.

We close by mentioning three relevant directions for future research: *i) Conditional meta-learning* Is it possible to extend the framework to conditional meta-learning? One way would be to use KTA similar to [Sinha and Duchi, 2016] and adjusting the initial starting kernel similar to MAML; *ii) Theoretical guarantees* Can we show that IKML converges to a stationary point for benign settings? This would require understanding the bias-variance decomposition of the gradient; *iii) Alternative Kernel Classes* Can we extend IKML to other kernel families? An example is dot-product kernels [Kar and Karnick, 2012].

# 7 ACKNOWLEDGEMENTS

The authors would like to thank comments by the anonymous reviewers which helped improve the paper. John Isak Texas Falk is supported through the Department of Computer Science at UCL, and this work was partially carried out while he was a visiting doctoral student at Istituto Italiano di

Tecnologia, Genoa. Carlo Ciliberto acknowledges the support of the Royal Society (grant SPREM RGS\R1\201149) and Amazon.com Inc. (Amazon Research Award – ARA).

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
