# OpenReview forum: "Implicit kernel meta-learning using kernel integral forms"
_auai.org/UAI/2022/Conference — UAI 2022 Oral_

### Official Review · Reviewer_L9WH · 2022-04-15

**Q2(1) Originality/Novelty:** 2
**Q2(2) Significance/Impact:** 2
**Q2(3) Correctness/Technical Quality:** 3
**Q2(6) Clarity Of Writing:** 3
**Q6 Overall Score:** 5
**Q8 Confidence In Your Score:** 3

**Q1 Summary And Contributions:**

The paper proposes a new learning framework for meta-learning regression problem. Its inner algorithm relies on kernel ridge regression associated with a random feature kernel. The authors derive the generalization error bound in terms of the number of tasks, the task sample size and the number of random features. The proposed algorithms are also illustrated by several numerical experiments.

**Q2 Assessment Of The Paper:**

More detailed information regarding each of these aspects is given below:

**Q2(4) Quality Of Experiments (Optional):**

2: Fair: The experimental evaluation is weak: important baselines are missing, or the results do not adequately support the main claims.

**Q2(5) Reproducibility:**

3: Good: Key resources (e.g., proofs, code, data) are available and key details (e.g., proofs, experimental setup) are sufficiently well-described for competent researchers to confidently reproduce the main results.

**Q3 Main Strengths:**

Random feature model is considered in the framework of kernel meta-learning. Generalization error bound of the proposal algorithm is provided.

**Q4 Main Weakness:**

I cannot find any assumption for Theorem 1, which seems to be weird. Claims on the performance in the numerical experiments are also not convincing.

**Q5 Detailed Comments To The Authors:**

1). In Theorem 1, is there any requirement for the step size $\gamma_t$ to guarantee the convergence of the proposed algorithm?

2). In Figure 1, what does the figure for d=20 imply? It is not a too high dimension.

3). In Table 1, the performance of the proposed algorithms seem to be not significantly better than R2D2, then what is the main benefit of the proposed method, computational efficiency?

**Q7 Justification For Your Score:**

It is novel and interesting to introduce random feature model to kernel meta-learning regression problems, but some further clarification would be necessary.

**Q9 Complying With Reviewing Instructions:**

1: Yes.

---

### Official Review · Reviewer_Yf3H · 2022-04-16

**Q2(1) Originality/Novelty:** 3
**Q2(2) Significance/Impact:** 2
**Q2(3) Correctness/Technical Quality:** 3
**Q2(6) Clarity Of Writing:** 3
**Q6 Overall Score:** 7
**Q8 Confidence In Your Score:** 3

**Q1 Summary And Contributions:**

The authors mainly propose the framework called implict kernel meta-learning (IKML) that implicitly learns the representation (e.g., meta paramether $\theta$) via a kernel function. They use translation-invariant kernel, especially kernel ridge regression, which can be formulated as integral forms and be approximated by random feature sampling. They also derive the generalization bound for IKML.

**Q2 Assessment Of The Paper:**

More detailed information regarding each of these aspects is given below:

**Q2(5) Reproducibility:**

4: Excellent: Key resources (e.g., proofs, code, data) are available and key details (e.g., proof sketches, experimental setup) are comprehensively described for competent researchers to confidently and easily reproduce the main results.

**Q3 Main Strengths:**

•	The motivation and the contribution is clear.
•	They demonstrate the proposed algorithm and corresponding theorem elaborately.
•	On the experiment part, they describe in detail so that the other researchers can reproduce the results.


**Q4 Main Weakness:**

•	The experiment results are quite insufficient. The visualization of experiment results should be added. Furthermore, assessing the methods only with RMSE doesn’t seem sound enough. I think using additional metric would be good for the performance justificaiton.
•	It seems that explanatory part for the various concepts and theorem used in the paper is needed. Though the references are well-written, I think it would be easier to understand if all related concepts are organized in one place.


**Q5 Detailed Comments To The Authors:**

•	On Example 1 (Affine Pushforward Maps), it seems there is a typo. $K_{\tau(M,b)}$ may be changed into $K_{\tau(Q,b)}$.
•	It would be better if there are some explanations regarding the criteria of choosing baselines on each experiment.
•	I think additional metric besides RMSE is necessary to guarantee the perform of IKML.
•	Why do you use different depth of MLP on MAML and IKML in “air quality experiment”?


**Q7 Justification For Your Score:**

I gave score 7 (Accept) because of following reasons. The motivation and the contirbution of paper are clear. To be specific, the proposed framework (IKML) is justified on logical way and its theoretical guarantees has been well proven. Also, the experiment results support the the performance of their algorithm.

**Q9 Complying With Reviewing Instructions:**

1: Yes.

---

### Official Review · Reviewer_VAb8 · 2022-04-23

**Q2(1) Originality/Novelty:** 3
**Q2(2) Significance/Impact:** 3
**Q2(3) Correctness/Technical Quality:** 3
**Q2(6) Clarity Of Writing:** 3
**Q6 Overall Score:** 6
**Q8 Confidence In Your Score:** 5

**Q1 Summary And Contributions:**

This paper presents a framework for implicit kernel meta-learning (IKML) in context of translation-invariant and deep
random kernel families.The core idea is to explore problems where data does not present a clear input structure and using a plug-in
translation invariant kernel might be a safer strategy. For this purpose, this paper derive a novel bound on the excess transfer risk and perform an empirical study on two real-world meta-learning regression datasets.


**Q2 Assessment Of The Paper:**

More detailed information regarding each of these aspects is given below:

**Q2(4) Quality Of Experiments (Optional):**

2: Fair: The experimental evaluation is weak: important baselines are missing, or the results do not adequately support the main claims.

**Q2(5) Reproducibility:**

3: Good: Key resources (e.g., proofs, code, data) are available and key details (e.g., proofs, experimental setup) are sufficiently well-described for competent researchers to confidently reproduce the main results.

**Q3 Main Strengths:**

- The paper is well-written.
- The theoretical work of this paper is sufficient, which improves the value of the paper.
- The paper has several novelties: i) This paper introduces two instances of this meta-learning framework, learning a neural network pushforward for a translation-invariant kernel and an affine pushforward for a neural network random feature kernel, both mapping from a Gaussian latent distribution; ii) For this purpose, The authors adopt a random feature sampling approach to approximate it an analytical form，which is called as Implicit Kernel Meta-Learning (IKML); iii) This paper demonstrate the effectiveness in an extensive empirical study and theoretical analysis.


**Q4 Main Weakness:**

- The experimental analysis is weak, in that there are few comparisons against prior state of the art methods. Since most comparative methods are obsolete，more recently algorithm such as [A,B] should be considered. Besides, the performance of the method on the Gas Sensor dataset is not significant, which makes me doubt the effectiveness of the method.
- Although the theoretical work of this paper is sufficient， some writing content confuses me. The authors only list Proof Sketch in the main text but does not deeply analyze the meaning of the formula.

[A] Kong W, Somani R, Kakade S, et al. Robust meta-learning for mixed linear regression with small batches. Advances in neural information processing systems, 2020, 33: 4683-4696
[B] Timothy Nguyen and Zhourong Chen and Jaehoon Lee,Dataset Meta-Learning from Kernel Ridge-Regression,International Conference on Learning Representations,2021.



**Q5 Detailed Comments To The Authors:**

Can the author give the algorithm complexity and the memory overhead to compare the MAML approach?

**Q7 Justification For Your Score:**

Please see Q3 and Q4

**Q9 Complying With Reviewing Instructions:**

1: Yes.

---

### Decision · Program_Chairs · 2022-05-15

**Decision:**

Accept (Oral)

**Comment:**

Meta Review: I recommend to accept this paper.

This paper proposes a framework for implicit kernel meta-learning (IKML) in context of translation-invariant and deep random kernel families. The core idea is to explore problems where data does not present a clear input structure and using a plug-in translation invariant kernel might be a safer strategy. For this purpose, this paper derives a novel bound on the excess transfer risk and performs an empirical study on two real-world meta-learning regression datasets. Reviewers point out that the paper is well-written and organized, the motivation is clear, the theoretical work is sufficient, and the authors demonstrate the proposed algorithm elaborately. The authors handle the questions well in the rebuttal. In summary, reviews generally agree that the paper is novel and technically well written. The reviewers also make some useful comments that should be reflected in the final version. The authors should polish the experimental section.